# Copper Is Accumulated as Copper Sulfide Particles, and Not Bound to Glutathione, Phytochelatins or Metallothioneins, in the Marine Alga *Ulva compressa* (Chlorophyta)

**DOI:** 10.3390/ijms25147632

**Published:** 2024-07-11

**Authors:** Stephanie Romero, Alberto González, Héctor Osorio, Rodrigo Segura, Alejandra Moenne

**Affiliations:** 1Laboratory of Marine Biotechnology, Faculty of Chemistry and Biology, University of Santiago of Chile, Alameda 3363, Santiago 917022, Chile; stephanie.romero@usach.cl (S.R.); alberto.gonzalezfi@usach.cl (A.G.); hector.osorio@usach.cl (H.O.); 2Laboratory of Inorganic Chemistry, Faculty of Chemistry and Biology, University of Santiago of Chile, Alameda 3363, Santiago 917022, Chile; rodrigo.segura@usach.cl

**Keywords:** copper, electrondense particles, glutathione, metallothioneins, phytochelatins, sulfide

## Abstract

To analyze the mechanism of copper accumulation in the marine alga *Ulva compressa*, it was cultivated with 10 μM of copper, with 10 μM of copper and increasing concentrations of a sulfide donor (NaHS) for 0 to 7 days, and with 10 μM of copper and a concentration of the sulfide acceptor (hypotaurine) for 5 days. The level of intracellular copper was determined as well as the level of glutathione (GSH) and phytochelatins (PCs) and the expression of metallothioneins (UcMTs). The level of intracellular copper in the algae treated with copper increased at day 1, slightly increased until day 5 and remained unchanged until day 7. The level of copper in the algae cultivated with copper and 100 or 200 μM of NaHS continuously increased until day 7 and the copper level was higher in the algae cultivated with 200 μM of NaHS compared to 100 μM of NaHS. In contrast, the level of intracellular copper decreased in the algae treated with copper and hypotaurine. The level of intracellular copper did not correlate with the level of GSH or with the expression of UcMTs, and PCs were not detected in response to copper, or copper and NaHS. Algae treated with copper and with copper and 200 μM of NaHS for 5 days were visualized by TEM and the elemental composition of electrondense particles was analyzed by EDXS. The algae treated with copper showed electrondense particles containing copper and sulfur, but not nitrogen, and they were mainly located in the chloroplast, but also in the cytoplasm. The algae treated with copper and NaHS showed a higher level of electrondense particles containing copper and sulfur, but not nitrogen, and they were located in the chloroplast, and in the cytoplasm. Thus, copper is accumulated as copper sulfide insoluble particles, and not bound to GSH, PCs or UcMTs, in the marine alga *U. compressa*.

## 1. Introduction

Copper is an essential heavy metal that is required as a cofactor for the activity of several enzymes and proteins such as Cu/Zn superoxide dismutase, laccase, ascorbate oxidase, plastoquinone, cytochrome c and ethylene receptor, among others [1,2]. In contrast, cadmium is a non-essential heavy metal, since it does not act as a cofactor in proteins or enzymes, with the exception of carbonic anhydrase in a marine diatom that uses cadmium as a cofactor [3].

In bacteria, microalgae and some fungi, copper and cadmium are accumulated mainly as copper sulfide [4,5]. For example, copper is accumulated as copper sulfide in soil bacteria and sulfide is produced by enzymes such as L-cysteine desulfhydrase (L-CDS), cystathionine lyase (CL), cystathionine synthase (CS) and 3-mercaptopyruvate sulfurtransferase (MPST), mainly by L-CDS and MPST [5]. In the green microalga *Chlamydomonas reinhardtii*, the red microalga *Cyanidioshyzon merolae* and the cyanobacteria *Synechoccocus leopoliensis*, cadmium is also accumulated as cadmium sulfide and L-CDS activity was found to be increased [4]. Interestingly, the mushroom *Agaricus brasiliensis* when exposed to cadmium showed an increase in in expression of cysteine synthase, L-CDS, CS and MPST, suggesting that the metal is accumulated as cadmium sulfide [6]. Thus, the accumulation of copper and cadmium as insoluble sulfur particles may represent an ancient mechanism of heavy metal accumulation, since it is observed in bacteria, microalgae and some fungi.

In the yeast *Saccharomyces cerevisiae*, in the flagellated green microalga *Euglena gracilis* and in terrestrial plants, copper and cadmium are accumulated mainly through their binding to glutathione (GSH), a tripeptide constituted by cysteine, glutamate and glycine; to phytochelatins (PCs) that are peptides formed by condensation of GSH and synthesized by the enzyme phytochelatin synthase (PCS) and/or to metallothioneins (MTs) that are gene-encoded small proteins rich in cysteines [2,7]. In plants, GSH, PCs and MTs bind copper and cadmium ions through sulfhydryl groups of cysteines. In the yeast *S. cerevisiae*, copper is accumulated bound to two GSH in the vacuole [8]. In *E. gracilis*, copper is accumulated complexed with GSH and sulfide in the chloroplasts [9]. Regarding the involvement of PCs in heavy metal accumulation in plants, the overexpression of PCS from the aquatic macrophyte *Ceratophyllum demersum* in tobacco plants allowed a higher accumulation of cadmium and arsenic compared to control plants [10]. In addition, the overexpression of PCS of the herb *Boehmeria nivea* in *Arabidopsis thaliana* allowed higher accumulation of cadmium compared to control plants [11]. Moreover, the mutation of PCS in the bryophyte *Marchantia polymorpha* induced sensitivity to cadmium, but not to other metals, and its overexpression allowed higher accumulation of cadmium compared to control plants, indicating that the major role of PCs is cadmium detoxification in lower plants [12]. Regarding expression of MTs and heavy metal accumulation in plants, five MTs of *A. thaliana* expressed in *S. cerevisiae* that lacked CUP-1 MT allowed the accumulation of copper [13]. Furthermore, the overexpression of a *Brassica napus* MT in *S. cerevisiae*, but not two other BnMTs, allowed the accumulation of cadmium [14]. In addition, an *A. thaliana* mutant lacking two MT genes accumulated 30% less copper and that lacking four MT genes accumulated 45% less copper than the double mutant [15], suggesting that plant MTs may be involved in copper accumulation. Thus, GSH, PCs and MTs are involved in copper and cadmium accumulation in terrestrial plants *S. cerevisiae* and *E. gracilis*. 

The marine macroalga *Ulva compressa* is a cosmopolitan green macroalga that is the dominant species in costal sites contaminated with copper by receiving effluents of copper mines in northern Chile [16]. *U. compressa* from copper-contaminated sites accumulated copper in its tissue and showed synthesis of ascorbate, accumulated as dehydroascorbate, and activation of the antioxidant enzyme ascorbate peroxidase [16]. In addition, *U. compressa* cultivated in vitro with increasing concentrations of copper (2.5 to 10 μM of copper) for 0 to 12 days also accumulated copper in its tissue and the intracellular accumulation linearly increased with increasing concentrations of copper and with time [17]. The increasing concentrations of copper induced the increase in GSH and PCs levels as well as in the expression of three *U. compressa* MTs designated as UcMT1.1, UcMT2 and UcMT3 [17,18]. The transcripts encoding these UcMTs were cloned and expressed in *Escherichia coli,* allowing the accumulation of copper and zinc in the bacteria, suggesting that they may participate in copper accumulation in the alga [18]. In addition, the genome of *U. compressa* was sequenced, and another three new UcMTs were detected, showing that their amino acid sequence is closely related to UcMT1.1, and this UcMT1.1 family includes UcMT1.2, UcMT1.3 and UcMT1.4 [19], see Figure 1.

In this work, *U. compressa* was cultivated with 10 μM of copper (control), with 10 μM of copper and increasing concentrations of NaHS, a sulfide donor, for 0 to 7 days and with a concentration of hypotaurine, a sulfide acceptor, for 5 days. The level of intracellular copper was determined as well as the content of GSH and PCs and the expression of the six UcMTs. In addition, the accumulation of intracellular copper was analyzed by transmission electron microscopy (TEM) coupled to emission dispersive X-ray spectroscopy (EDXS) in the alga treated with 10 μM of copper, and with 10 μM of copper and 200 μM of NaHS for 5 days, to determine the amounts of electrondense particles, their cellular location and their elemental composition. The results demonstrated that copper was accumulated as copper sulfide particles, mainly in the chloroplast, which may correspond to an ancient mechanism observed in bacteria, microalgae and some fungi, but not in terrestrial plants.

## 2. Results 

### 2.1. U. compressa Metallothioneins (UcMTs)

The UcMT1.1 family is constituted by UcMT1.1, UcMT1.2, UcMT1.3 and UcMT1.4. UcMT1.1 contains 81 amino acids; its molecular weight (MW) is 8.2 kDa and presents 25 cysteines, which corresponds to 25% of total amino acids. UcMT1.2 is constituted by 77 amino acids; its MW is 7.5 kDa and contains 20 cysteines, which corresponds to 26% of total amino acids. UcMT1.3 is constituted by 89 amino acids; its MW is 8.7 kDa and contains 25 cysteines, which corresponds to 28% of total amino acids. UcMT1.4 is constituted by 87 amino acids; its MW is 8.6 kDa and it contains 26 cysteines, which corresponds to 30% of total amino acids (Figure 1, Table 1).

In contrast, UcMT2 is constituted by 90 amino acids; its MW is 9.1 kDa and it contains 27 cysteines, which corresponds to 30% of total amino acids. UcMT3 is constituted by 139 amino acids; its MW is 13.4 kDa and it contains 34 cysteines, which corresponds to 24% of total amino acids (Figure 1; Table 1). Thus, the UcMT1.1 family has a MW from 7.5 to 8.7 kDa and contains 25–30% of cysteines residues, whereas UcMT2 and UcMT3 have a higher MW than the UcMT1.1 family of 9.3 and 13.4 kDa, respectively, and a similar cysteine content of 30 and 24%, respectively. 

### 2.2. Intracellular Accumulation of Copper in Response to Copper, NaHS and Hypotaurine 

To analyze the effect of NaHS, a sulfide donor, on intracellular copper accumulation in *U. compressa*, adult algae collected in southern Chile were cultivated with 10 μM of copper, and with 10 μM of copper and 100 μM of NaHS or 200 μM of NaHS for 0 to 7 days. The level of copper in the algae treated with 10 μM of copper was 57 μg g^−1^ of DT at day 1, 72 μg g^−1^ of DT at day 3 and 80 μg g^−1^ of DT at days 5 and 7 (Figure 2A). Intracellular copper in the algae treated with copper and 100 μM of NaHS was 29 μg g^−1^ of DT at day 1, 58 μg g^−1^ of DT at day 3, 93 μg g^−1^ of DT at day 5 and 111 μg g^−1^ of DT at day 7, which represents an increase of 54.5% compared to the copper level in the algae treated only with copper at day 7 (Figure 2A). The level of copper in the algae treated with copper and 200 μM of NaHS was 32 μg g^−1^ of DT at day 1, 54 μg g^−1^ of DT at day 3, 108 μg g^−1^ of DT at day 5 and 138 μg g^−1^ of DT at day 7, which represents an increase of 98.6% compared to the copper level in the algae treated only with copper at day 7 (Figure 2A). Thus, an increase in the amount of a sulfide donor enhanced the accumulation of copper, suggesting this metal is accumulated as copper sulfide in the alga. 

To analyze the effect of hypotaurine, a sulfide acceptor, on *U. compressa*, juvenile algae collected in southern Chile were cultivated with 10 μM of copper and with 10 μM of copper and 500 μM of hypotaurine, a sulfide acceptor, for 5 days (Figure 2B). The level of copper in the algae treated with copper was 137 μg g^−1^ of DT and in the algae treated with copper and hypotaurine it was 87 μg g^−1^ of DT, which represents a decrease of 36.5% (Figure 2B). Thus, the level of intracellular copper decreased in response to a sulfide acceptor, also suggesting that copper is accumulated as copper sulfide in the alga.

### 2.3. Level of GSH and PCs in Response to Copper and NaHS

To analyze whether GHS and/or PCs are involved in intracellular copper accumulation in *U. compressa*, adult algae collected in southern Chile were cultivated with 10 μM copper and with 10 μM of copper and 100 μM or 200 μM of NaHS for 0 to 7 days. The level of GSH in the alga cultivated with 10 μM of copper was 246 μg g^−1^ of DT and it did not significantly change until day 7 (Figure 3). The level of GSH in the alga treated copper and 100 μM of NaHS was 317 μg g^−1^ of DT at day 1, decreased to 276 μg g^−1^ of DT at day 3, increased to 324 μg g^−1^ of DT at day 5, and further increased to 340 μg g^−1^ of DT at day 7, which represents an increase of 38.2% compare to GSH in the algae treated only with copper at day 7 (Figure 3). In the algae treated with copper and 200 μM of NaHS, the level of GSH was 294 μg g^−1^ of DT at day 1, decreased to 285 μg g^−1^ of DT at day 3, increased to 331 μg g^−1^ of DT at day 5 and increased again to 353 μg g^−1^ of DT at day 7, which represents an increase of 43.5% compared to GSH in the algae treated with copper at day 7 (Figure 3). Thus, the level of GSH in the algae treated with copper and with two concentrations of NaHS was similar at day 7, suggesting that GSH is not involved in copper accumulation in the alga. Interestingly, PCs were not detected in the algae treated with copper or with copper and 100 μM or 200 μM of NaHS from 0 to 7 d, which indicates PCs are not involved in copper accumulation in the alga. 

### 2.4. Expression of UcMT1.1 Family in Response to Copper and NaHS

To analyze whether UcMTs are involved in intracellular copper accumulation in *U. compressa*, adult algae collected in southern Chile were cultivated without copper, with 10 μM copper and with 10 μM of copper and 50, 100 or 200 μM of NaHS for 0 to 7 days and the relative level of UcMT transcripts was determined. The relative level of transcripts encoding UcMT1.1 increased 2 times at day 1, remained unchanged until day 5 and increased 8 times at day 7 in response to 10 μM of copper (Figure 4A); transcripts increased 11, 4 and 3 times at days 1, 3 and 5, respectively, and then decreased to control level at day 7 in response to 10 μM of copper and 50 μM of NaHS (Figure 4B); transcripts increased 5 and 8 times at days 5 and 7, respectively, in response to copper and 100 μM of NaHS (Figure 4C). Surprisingly, transcripts of UcMT1.1 did not increase in response to 10 μM of copper and 200 μM of NaHS. Thus, UcMT1.1 may not be involved in copper accumulation in the alga, since the higher accumulation of copper was observed in response to copper and 200 μM of NaHS at day 7 and this concentration of NaHS did not increase the expression of MT1.1. 

The relative level of transcripts encoding UcMT1.2 increased 3 and 10 times at days 5 and 7, respectively, in response to 10 μM of copper (Figure 4D); transcripts increased 31, 40 and 63 times at days 1, 3 and 5, respectively, and decreased to control level at day 7 in response to 10 μM of copper and 50 μM NaHS (Figure 4E); transcripts increased 27, 53 and 80 times at days 3, 5 and 7 in response to 10 μM of copper and 100 μM NaHS (Figure 4F) and transcripts of UcMT1.2 did not increase in response to 10 μM of copper and 200 μM NaHS. Thus, UcMT1.2 is not involved in copper accumulation, since the highest accumulation of copper was observed in response to 10 μM of copper and 200 μM of NaHS at day 7 and this concentration of NaHS did not increase the expression of UcMT1.2. Interestingly, the increase in expression of UcMT1.2 was much higher than that of MT1.1 and the other UcMTs.

The relative level of transcripts encoding UcMT1.3 increased 3, 3, 2 and 3 times at days 1,3, 5 and 7 d, respectively, in response to 10 μM of copper (Figure 4G); transcripts increased 13, 8 and 5 times at days 1, 3 and 5, respectively, and decreased to control level at day 7 in response to 10 μM of copper and 50 μM NaHS (Figure 4H); transcripts increased 3 times at day 3, decreased to control level at day 5 and increased again 3 times at day 7 in response to 10 μM of copper and 100 μM NaHS (Figure 4I), and transcripts of UcMT1.3 did not increase in response to 10 μM of copper and 200 μM NaHS. Thus, UcMT1.3 is not involved in copper accumulation since the higher copper accumulation of copper was observed in response to copper and 200 μM of NaHS at day 7 and this concentration of NaHS did not increase the expression of MT1.3. 

The relative level of UcMT1.4 increased 3 times at days 1, 3 and 7 in response to 10 μM of copper (Figure 4J), it increased 7 and 5 times at days 1 and 3, respectively, and then decreased to control level at day 7 in response to 10 μM of copper 50 μM NaHS (Figure 4K); the level of transcripts increased 2 times at day 3 and then decreased to control level at day 7 in response to 10 μM of copper and 100 μM NaHS (Figure 4L), and transcripts of UcMT1.4 did not increase in response to 10 μM of copper and 200 μM NaHS. Thus, UcMT1.4 is not involved in copper accumulation since the higher copper accumulation of copper was observed in response to 10 μM of copper and 200 μM of NaHS at day 7 and this concentration of NaHS did not increase expression of MT1.4.

### 2.5. Expression of UcMT2 and UcMT3 in Response to Copper and NaHS 

The relative level of transcripts encoding UcMT2 increased 13 times at day 7 in response to 10 μM of copper (Figure 5A); transcripts increased 48 times at day 7 in response to 10 μM of copper and 50 μM NaHS (Figure 5B); transcripts increased 4 and 5 times at days 1 and 3, respectively, decreased to control level at day 3 and increased again 5 times at day 7 in response to 10 μM of copper and 100 μM NaHS (Figure 5C); transcripts increased 46 and 36 times at days 1 and 3, respectively, decreased at day 5 and increased again 38 times at day 7 in response to 10 μM of copper and 200 μM NaHS (Figure 5D). Thus, the increase in expression of UcMT2 differed from that observed for UcMT1.1, UcMT1.2, UCMT1.3 and UcMT1.4, since it increased at day 7 in response to 10 μM of copper and 200 μM NaHS, suggesting that it could be involved in copper accumulation in the alga. 

In addition, the level of transcripts encoding UcMT3 increased 2, 3, 2 and 6 times at days 1, 3, 5 and 7, respectively, in response to 10 μM of copper (Figure 5E) but did not increase in response to 10 μM of copper and 50, 100 or 200 μM of NaHS. Thus, UcMT3 is not involved in copper accumulation, since its expression did not increase in response to NaHS. Therefore, UcMTs may not be involved in copper accumulation, with the possible exception of UcMT2. 

### 2.6. Accumulation of Intracellular Copper as Copper Sulfide Particles 

The alga was cultivated with 10 μM of copper (control) and with 10 μM of copper with 200 μM of NaHS for 5 days. Control and treated algae were visualized by TEM and elements present in electrondense particles were analyzed by EDXS (Figure 6A–D), in triplicate. Control cells showed electrondense particles containing copper and sulfide, but not N (Appendix A), mainly located in the chloroplast, but also in the cytoplasm (Figure 6A,B). Furthermore, cells treated with 10 μM and 200 μM of NaHS showed higher amounts of electrondense particles compared to the alga cultivated only with copper. These particles contained copper and sulfide, but not N, (Appendix A) and they were located mainly in the chloroplast and the cytoplasm. Control cells showed a mean value of 2 electrondense particles containing copper in the cytoplasm and 7 electrondense particles located in the chloroplast. Cells treated with copper and NaHS showed a mean value of 8 electrondense particles in the cytoplasm and 19 electrondense particles in the chloroplast (Figure 7). In conclusion, treatment with copper and a sulfide donor (NaHS) increased the number of electrodense particles containing copper and sulfur, but not N, indicating that copper is accumulated as copper sulfide, and not bound to GSH, PCs or UcMTs, in *U. compressa*. 

## 3. Discussion

In this work, we showed that increasing concentrations of NaHS, a sulfide donor, increased the level of intracellular copper, and a concentration hypotaurine, a sulfide acceptor, decreased the level of the metal in *U. compressa,* suggesting that this metal is accumulated as copper sulfide. These results contrast with those described for copper accumulation in plants involving the binding of metals to GSH, PCs and/or MTs, and their transport to the vacuole [2]. However, these results are in accord with those obtained in bacteria, microalgae and some fungi that accumulate copper and other heavy metals as metal sulfide insoluble particles [4,5]. The latter is in accord with the finding that several protein-coding genes in *U. compressa* genome have homology with bacterial and microalgae genes that may allow mechanisms present in bacteria to occur in a marine macroalgae [19]. Thus, copper accumulation in marine macroalgae may be achieved through an ancient mechanism corresponding to metal accumulation as insoluble sulfide particles. 

It is important to mention that copper accumulation in adult algae collected in southern Chile and cultivated with 10 μM of copper reached a maximal level at day 5 that remained until day 7. This contrasts with previous results obtained with adult algae from central Chile showing a linear increase of intracellular copper until day 12 [17]. It is important to point out that *U. compressa* collected in central and southern Chile had identical morphologies but the kinetic of copper accumulation differed. In addition, adult algae collected in southern Chile cultivated with 10 μM of copper accumulated a lower amount of intracellular copper (80 μg g^−1^ of DT) compared to juvenile algae collected in this place (138 μg g^−1^ of DT). These inconsistencies may be due to algae in different developmental stages and/or to the existence of algal ecotypes. Furthermore, juvenile algae may capture copper from the culture medium faster than adult algae, since they may express higher level of copper transporters, but the latter need to be analyzed. A molecular characterization of the marine alga *U. compressa* will be performed in the future to determine whether the algae from central and southern Chile correspond to *U. compressa* ecotypes. 

In addition, the increase in GSH did not correlate with the increase in copper accumulation, since a similar increase in GSH level was observed in response to copper and 100 or 200 μM of NaHS at day 7, which contrasted with the higher increase in copper level in response to copper and NaHS compared to copper without NaHS at day 7. Thus, GSH may not be involved in copper accumulation in *U. compressa*. In addition, PCs were not detected in response to copper, or with copper and NaHS, indicating that PCs are not involved in the accumulation of copper in the alga. This contrasts with the results obtained in *U. compressa* collected in central Chile, which synthesized PCs in response to 10 μM of copper [17], also suggesting that *U. compressa* from central and southern Chile are algal ecotypes. These results also contrast with those obtained in plants such as alfalfa seedling cultivated with cadmium and sulfate as an S donor, showing that the level of GSH and PCs increased in response to an S donor and that cadmium was mainly accumulated in the roots bound to the cell wall, and in the vacuole [20]. In addition, cadmium increased the level of hydrogen sulfide, cysteine, GSH and PCs in the roots of *A. thaliana* as well as cadmium tolerance and the expression of PCS enzyme and a metallothionein [21]. Thus, GSH and PCs are not involved in copper accumulation in *U. compressa*, but they are involved in cadmium tolerance and accumulation in terrestrial plants.

On the other hand, the expression of UcMTs did not correlate with the increase in accumulation of copper inside the alga. In this sense, increasing concentrations of NaHS differentially regulate the expression of UcMTs, mainly 50 and 100 μM of NaHS, which increased the expression of the MT1.1 family, whereas 200 μM of NaHS inhibited their expression. In this case, it is possible that hydrogen sulfide was acting as a signaling molecule, as it has been shown to do in terrestrial plants [22,23], regulating the expression of UcMT1.1 family genes. In this sense, it has been shown that hydrogen sulfide increased tolerance to zinc in the Cd/Zn hyperaccumulator *Solanum nigrum* by decreasing oxidative stress through the activation of the antioxidant system and increasing the amount of the metallothionein SnMT1a [24]. In addition, *Brassica napus* treated with cadmium and NaHS showed a decrease in oxidative stress, an increased photosynthesis, activation of the antioxidant system and protection of membrane structures in organelles [25]. Interestingly, it has been shown that hydrogen sulfide acts on proteins through protein persulfidation and this modification may activate or inhibit the protein or the enzyme activity [26,27]. It has been shown that treatment with 50 μM of NaHS protects *A. thaliana* against salt stress and that mutants of JIN1/MYC2 transcription factor did not show protective effects, suggesting that hydrogen sulfide may act by persulfidation of this transcription factor [28]. Thus, it is possible that transcription factors in *U. compressa* may also be modified and activated by persulfidation, which may lead to an increased in the expression of UcMT1.1 family genes in response to 50 and 100 μM of NaHS. 

The most expressed of the UcMT1.1 family was UcMT1.2, showing a maximal increase of 62 times at day 5 in response to copper and 50 μM of NaHS, and an increase of 80 times at day 7 with copper and 100 μM of NaHS. In addition, UcMT2 showed an increased expression in response to 50, 100 and 200 M of NaHS. Thus, the UcMT1.1 family and UcMT2 showed an increase in their expression which may indicate that UcMTs are required in some way to mitigate the toxic effect of copper in *U. compressa*. In plants, it has been shown that MTs can also act as reactive oxygen species (ROS) scavengers, since rice OsMT1b overexpressed in yeast enable not only cadmium accumulation, but also hydrogen peroxide tolerance [29]. In addition, chromium II and VI induced the differential increased expression of 14 OsMTs in rice, leading to chromium tolerance, mainly OsMT1b, but also protected against hydrogen peroxide and superoxide anions, mainly OsMT2c [30]. Recently, it was shown that the C-terminal region of *Sedum alfredii* SaMT3 binds two cadmium ions whereas the N-terminal region binds ROS molecules such as hydrogen peroxide or superoxide anions [31]. Thus, MTs are involved in heavy metal chelation but also in ROS scavenging in plants, suggesting that the UcMTs that increased in response to copper may act as ROS scavengers in *U. compressa*. It is possible that ROS scavenging may represent the ancestral function of MTs in photosynthetic organisms. In conclusion, it was initially hypothesized that copper ions may bind to GSH, PCs and/or UcMTs in *U. compressa*, as in terrestrial plants *S. cereviciae* and *E. gracilis*, but here it was demonstrated that copper is accumulated as copper sulfide, as in bacteria, microalgae and fungi. 

It was previously shown that *U. compressa* from non-contaminated coastal sites cultivated with 25 μM of copper for 5 days and analyzed by TEM-EDXS showed electrondense particles containing copper and sulfur, and that they were located mainly in the chloroplast [32]. In this work, TEM-EDXS analyses of electrondense particles in the alga cultivated with 10 μM of copper for 5 d showed that copper is accumulated as particles containing copper and sulfide, but no N, and they were located mainly in the chloroplast, but also in the cytoplasm. In addition, the algae cultivated with 10 μM of copper and 200 μM of NaHS for 5 d showed an increased level of particles containing copper and sulfide, but no N, located in the chloroplast and in the cytoplasm. Thus, the sulfide donor NaHS increased level of copper sulfide particles in the chloroplast and the cytoplasm, indicating that copper accumulates as copper sulfide, and is not bound to GSH, PCS or UcMTs in *U. compressa,* which may represent an ancient mechanism of heavy metal accumulation. In the future, we will analyze whether other heavy metals such as cadmium and zinc may be accumulated as sulfide insoluble particles in *U. compressa*.

## 4. Materials and Methods

### 4.1. Sampling of U. compressa

Adults and juveniles of *U. compressa* were collected in Cocholgüe (36°36′ S) located in southern Chile, from September to December 2023, and this place showed a local temperature of 13–15 °C, seawater temperature of 12–13 °C and salinity of 34.8 g L^−1^. Algae were transported to the laboratory in a cooler, cleaned manually in synthetic seawater (30 g of sea salts in 1 L of distilled water) and sonicated twice for 2 min using an ultrasound bath (Hilab Innovation Systems, model SK221, Jilin, China) in order to remove epiphytic bacteria. Algae were maintained in aerated synthetic seawater (35 g of sea salts in 1 L of distilled water) in a culture chamber at 14 °C and with a light period of 12 h light/12 h darkness. 

### 4.2. Culture of U. compressa Treated with Copper, NaHS or Hypotaurine

Algae (10 g of fresh tissue [FT]) were cultivated in 300 mL of synthetic seawater without in copper (control), with 10 μM of copper, with 10 μM of copper and 100 or 200 μM of NaHS for 0, 1, 3, 5 and 7 days, or with 10 μM of copper and 500 μM of hypotaurine for 5 days. Algae were cultivated in aerated synthetic seawater in a culture chamber at 14 °C and with a light period of 12 h light/12 h darkness, in triplicate. After reaching culture time, algae were washed two times with 50 mM of Tris-HCl-10 mM EDTA to remove copper ions from the cell wall to quantify only intracellular copper. Algae (9 g of FT) were dried in an oven at 60 °C until reaching a constant weight, around 700–800 mg of dry tissue (DT) to quantify intracellular copper, GSH and PCs, and 1 g of FT to quantify UcMT transcripts. For detection of UcMTs transcripts, algae were also cultivated with 10 μM of copper and 50 μM of NaHS for 0 to 7 days. 

### 4.3. Quantification of Intracellular Copper in U. compressa

Algae (500 mg of dry weight (DT)) were incubated in ultrapure 9 mL of 65% nitric acid (ultrapure, Merck, MA, USA) and 1 mL of 30% hydrogen peroxide (Merck, MA, USA) in Teflon vials at 1800 W for 20 min to reach 210 °C, and for 15 min at 210 °C. When the temperature decreased to 30 °C, the vials were incubated at 4 °C for 20 min. The vials were opened, and the final volume was adjusted to 10 mL with distilled water. Quantification of intracellular copper was performed by atomic absorption spectroscopy (Analytic Jena, model Nova350, Jena, Germany) and the calibration curve was prepared with 0.5, 1, 2, 3, 5 and 7 μg mL^−1^ of copper. 

### 4.4. Quantification of GSH and PCs by HPLC

Algae (200 mg of DT) were frozen in liquid nitrogen and homogenized in a mortar using a pestle and 1.2 mL of 0.1% of trifluoroacetic acid (TFA)-6.3 mM dietilentriamine penta-acetic acid (DTPA) was added, and homogenization continued until thawing. The mixture was centrifuged at 14,000 rpm for 20 min and the supernatant was recovered. The supernatant was filtered using 0.22 μm PVDF membranes and an aliquot of 25 μL was mixed with 45 μL of 200 mM HEPES (pH = 8)-6.3 mM DTPA and 1 μL of 25 mM of monobromo-bimane and incubated at room temperature in darkness. The reaction was stopped by addition of 30 μL of 1 mM of methane-sulfonic acid.

The amount of GSH and PCs was determined using high-performance liquid chromatography (HPLC) equipment from Agilent Technologies, model 1260 Infinity (Santa Clara, CA, USA) and data were obtained using OpenLab software, version A.01.04.218. An aliquot of 20 μL was separated using a reverse phase C-18 column having a 5 μm particle size, 4.6 mm internal diameter and 15 cm length (Agilent Technologies, Santa Clara, CA, USA) at 24 °C, eluted using solvent A (0.1% TFA in aqueous solution) and solvent B (100% acetonitrile) and a linear gradient of 10 min from 0–20%, 30 min from 20–35% and 10 min from 35–100% of solvent B and a flow rate of 1 mL min^−1^. Pure GSH, PC2 and PC4 were dissolve in distilled water and used as standards. GSH and PCs were detected using a fluorescence detector and an excitation wavelength of 380 nm and an emission wavelength of 470 nm. Retention times of GSH, PC2 and PC4 were 9.25, 11.6 and 16.8 min, respectively, and the calibration curves were prepared using GSH, PC2 and PC4 at concentrations of 1 to 50 μM. 

### 4.5. Quantification of the Relative Levels of UcMT Transcripts by qRT-PCR

Algae (100 mg of FT) were frozen in liquid nitrogen and pulverized in a mortar with a pestle. Total RNAs were extracted using an EZNA total RNA kit (Omega Biotek, Norcross, GA, USA) and quantified by absorbance at 260 nm using an Infinite F500 microplate analyzer (Tecan, Zurich, Switzerland). The relative level of transcripts encoding UcMT1.1, UcMT1.2, UcMT1.3, UcMT1.4, UcMT2 and UcMT3, and transcripts encoding the housekeeping β-tubulin, were amplified with PCR primers described in Table 2. Synthesis of cDNAs was performed using 1 μg of total RNA, and iScript reverse transcription mix (BioRad, Hercules, CA, USA) completing to a final volume of 20 μL of water treated with DEPC (water-DEPC) and incubated at 25 °C for 5 min, at 46 °C for 20 min and at 95 °C for 1 min. Amplification of UcMTs and β-tubulin cDNAs was performed using 2 μL of cDNAs, 5 μL of SSO advanced SYBR green supermix (BioRad, CA, USA) and 10 μM of each PCR primer and completed to a final volume of 10 μL with water-DEPC. The amplification cycles were performed at 95 °C for 40 s, at 55 °C for 10 s and at 65 °C for 30 s using a thermocycler Aria MX (Agilent, CA, USA). 

### 4.6. Analysis of Copper Sulfide Nanoparticles by TEM-EDX

Algae (2 laminae) were introduced in 2 mL of 0.1 M cacodylate buffer (pH = 7.2) containing 1% glutaraldehyde and stored at 4 °C. Algae were included in epoxide resin, stained with osmium and uranyl, and ultrathin sections of 80 nm were obtained using an ultramicrotome (Leica, Wetzlar, Germany). Samples were placed on gold grids and analyzed using a TEM (Thermo Fisher Scientific, Waltham, MA, USA) model Talos of 200 KeV coupled to an EDX spectrometer model Quantax Xflash 6T-30 (Bruker, Coventry, UK) located at Center of Biomedical Research, University of Granada, Granada, Spain. 

### 4.7. Statistical Analyses

Significant differences were determined by two-way ANOVA followed by Tukey’s multiple comparison tests using the Shapiro–Wilk normality test and the software GraphPad Prism 8. Differences among mean values were considered to be significant at a probability of 5% (*p* < 0.05).

## Figures and Tables

**Figure 1 ijms-25-07632-f001:**
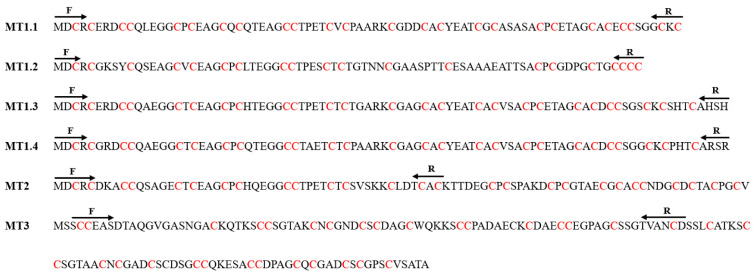
Amino acid sequences of *U. compressa* metallothioneins (UcMTs). Cysteine residues are depicted in red. Arrows indicate the positions of PCR primers used to amplify UcMT cDNAs.

**Figure 2 ijms-25-07632-f002:**
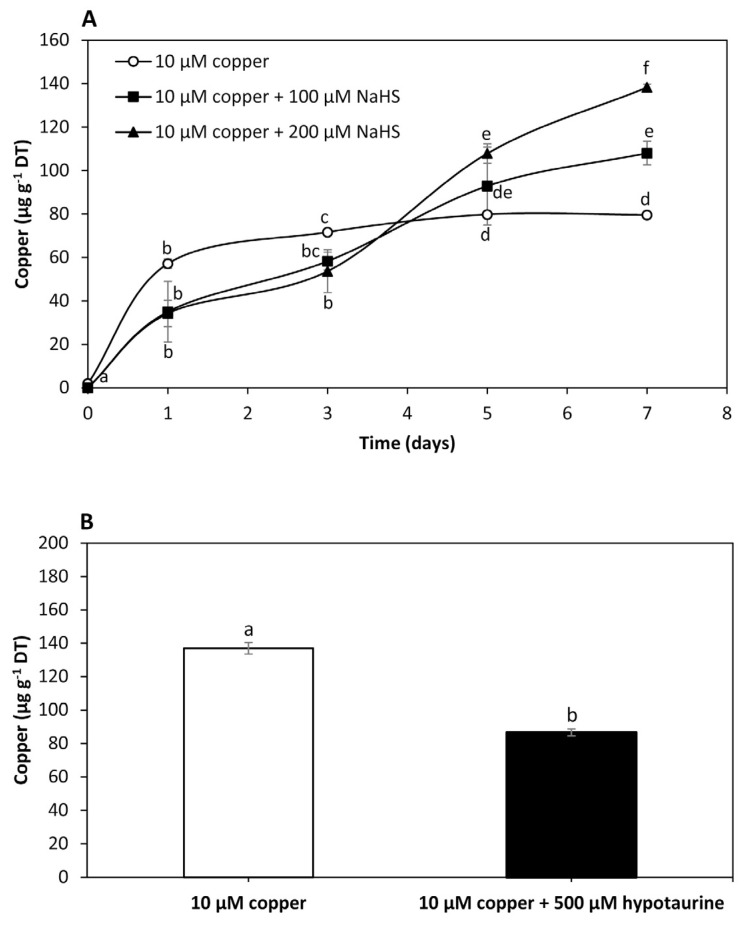
Level of intracellular copper in adult *U. compressa* algae treated with 10 μM of copper (open circles), 10 μM of copper and 100 μM NaHS (black squares) and with 10 μM of copper and 200 μM NaHS (black triangles) (**A**) for 0 to 7 days, and in juvenile algae treated with 10 μM of copper and 10 μM of copper and 500 μM of hypotaurine (**B**) for 5 days. The level of intracellular copper is expressed as micrograms per gram of dry tissue (DT). Symbols (**A**) and bars (**B**) represent mean values of three independent replicates ± SD. Letters indicate significant differences (*p* < 0.05).

**Figure 3 ijms-25-07632-f003:**
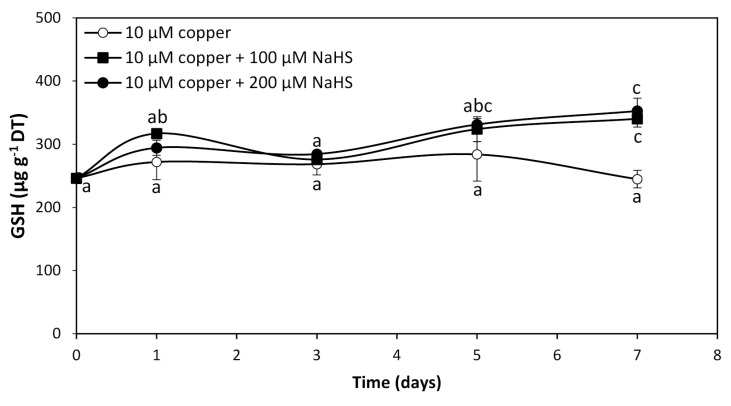
Level of reduced glutathione (GSH) in adult *U. compressa* algae treated with 10 μM of copper (open circles), 10 μM of copper and 100 μM NaHS (black squares) and with 10 μM of copper and 200 μM NaHS (black triangles) for 0 to 7 d. The level of GSH is expressed as micrograms per gram of dry tissue (DT). Symbols represent mean values of three independent replicates ± SD. Letters indicate significant differences (*p* < 0.05).

**Figure 4 ijms-25-07632-f004:**
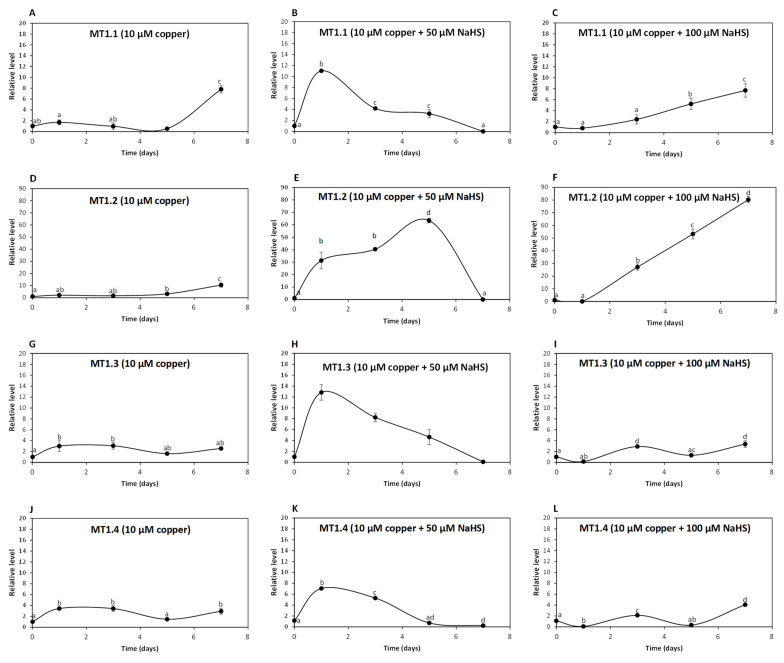
Relative level of transcripts encoding *U. compressa* metallothioneins (UcMTs) belonging to the MT1.1. family, corresponding to UcMT1.1., UcMT1.2, UcMT1.3 and UcMT1.4 in algae treated with 10 μM of copper (**A**,**D**,**G**,**J**), with 10 μM of copper and 50 μM of NaHS (**B**,**E**,**H**,**K**) and with 10 μM of copper and 100 μM of NaHS (**C**,**F**,**I**,**L**) for 0 to 7 d. The relative level of transcripts is expressed as 2^−ΔΔCT^. Symbols represent mean values of three independent replicates ± SD. Letters indicate significant differences (*p* < 0.05).

**Figure 5 ijms-25-07632-f005:**
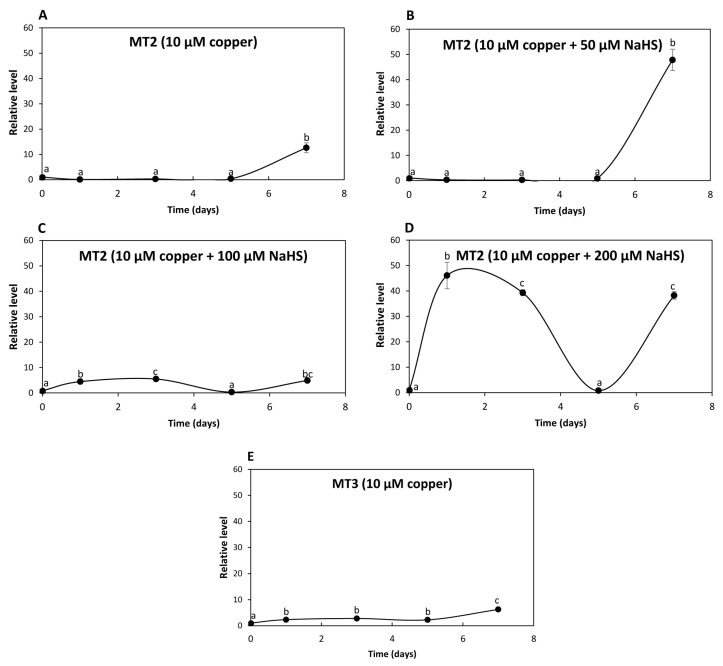
Relative level of transcripts encoding *U. compressa* metallothioneins UcMT2 in the algae cultivated with 10 μM of copper (**A**,**E**), with 10 μM of copper and 50 μM of NaHS (**B**), with 10 μM of copper and 100 μM of NaHS (**C**) and with 10 μM of copper and 200 μM of NaHS (**D**) for 0 to 7 d. Relative level of transcripts encoding UcMT3 in the alga cultivated with 10 μM of copper (**E**). The relative level of transcripts is expressed as 2^−ΔΔCT^. Symbols represent mean values of three independent replicates ± SD. Letters indicate significant differences (*p* < 0.05).

**Figure 6 ijms-25-07632-f006:**
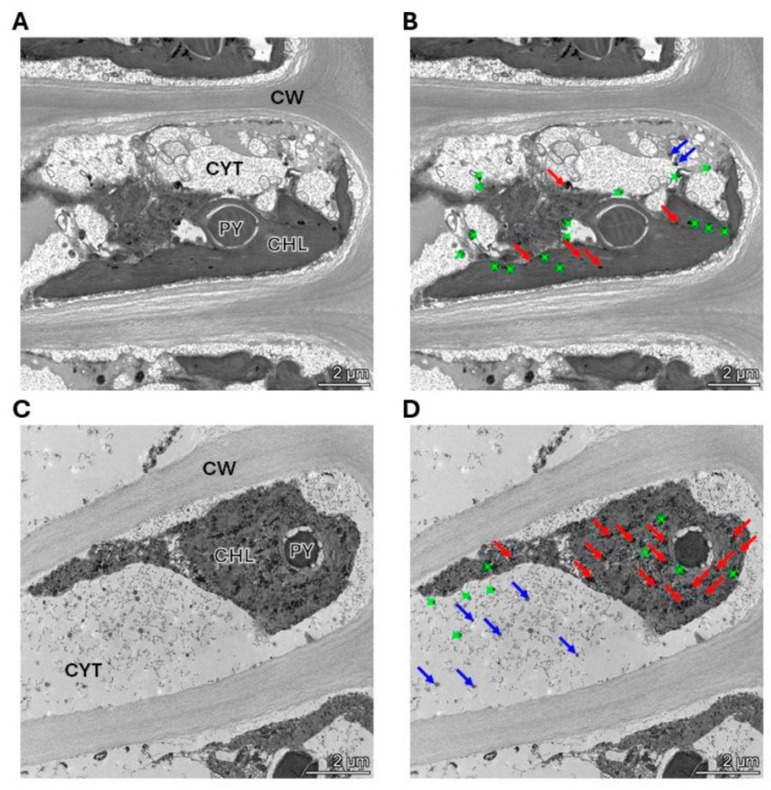
Visualization and analysis of electrondense particles by TEM-EDXS in a cell of the alga *U. compressa* cultivated with 10 μM of copper (**A**,**B**) and with 10 μM of copper and 200 μM of NaHS (**C**,**D**) for 5 days. In panels (**A**,**C**), CHL: chloroplast; PY: pyrenoid; CYT: cytoplasm and CW: cell wall. Panels (**A**,**B**) are the same picture as panels (**C**,**D**). Red arrows indicate copper sulfide particles in the chloroplast, blue arrows indicate copper sulfide in the cytoplasm and green dots indicate particles that do not contain copper.

**Figure 7 ijms-25-07632-f007:**
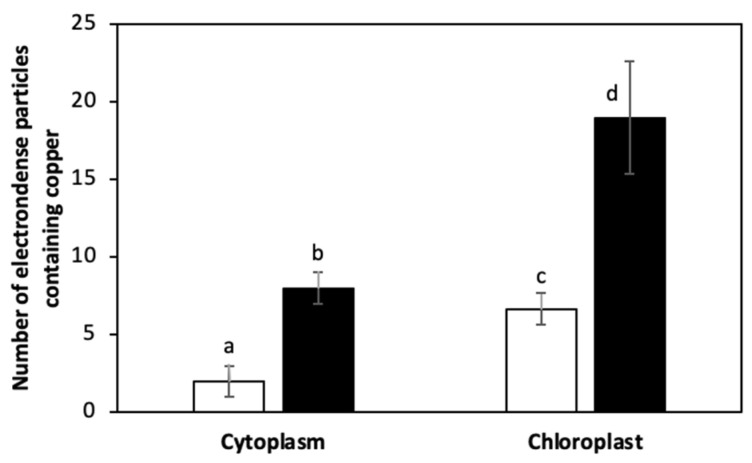
Number of electrondense particles containing copper in the cytoplasm and in the chloroplast of *U. compressa* cultivated with 10 μM of copper (open bars) and 10 μM of copper and 200 μM of NaHS (black bars) for 5 days. Bars represent mean values of three independent replicates ± SD. Letters indicate significant differences (*p* < 0.05).

**Table 1 ijms-25-07632-t001:** *U. compressa* metallothioneins.

Metallothioneins	Number of Amino Acids	Molecular Weight (kDa)	Number of Cysteines	Percentage of Cysteines
MT1.1	81	8.2	25	25
MT1.2	77	7.5	20	26
MT1.3	89	8.7	25	28
MT1.4	87	8.6	26	30
MT2	90	9.1	27	30
MT3	139	13.4	34	24

**Table 2 ijms-25-07632-t002:** PCR primers to amplify UcMT cDNAs.

Gene	Forward (F)	Reverse (R)
Tubulin	5′-TGCAACTTTTGTAGGCAACTC-3′	5′-CAGTGAACTCCATCTCGTCC-3′
MT1.1	5′-CATGGACTGCCGTTGCG-3′	5′-AGCTAGCACTTGCAACCG C-3′
MT1.2	5′-CATCATGGATTGCCGCTG-3′	5′-ATCAGCAGCAGCAGCAGC-3′
MT1.3	5′-CATGGACTGCCGTTGCG-3′	5′-AGCTAGTGCGAGTGAGCGC-3′
MT1.4	5′-CATGGACTGCCGTTGCG-3′	5′-CATCTAGCGCGAGCGAGC-3′
MT2	5′-ATGGACTGCCGTTGCGAC-3′	5′-TCTTGCAGGCGCAGGTG-3′
MT3	5′-TCTTGTTGTGAAGCCAGTGA-3′	5′-CACAGTTGCATTCTGCGGTT-3′

## Data Availability

https://doi.org/10.6084/m9.figshare.26236826.v1.

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
