# Peer review of "Copper Is Accumulated as Copper Sulfide Particles, and Not Bound to Glutathione, Phytochelatins or Metallothioneins, in the Marine Alga Ulva compressa (Chlorophyta)"

_ijms, 2024, doi:10.3390/ijms25147632_

Round 1

Reviewer 1 Report

Comments and Suggestions for Authors

1. There is a problem with the structure of the article, why the materials and methods are placed at the end please explain (usually after the introduction) and a conclusion should be added to summarize your findings.

2. The introduction needs to set out more clearly the background and significance of the study, which is currently too long and partly not directly related to the subject of the study.

3. Some of the results lack detailed data presentation and statistical analysis to adequately support the conclusions drawn, and in lines 285-295, while an increase in copper and sulfides is mentioned, no specific data points or statistical results are provided to demonstrate this change.

4.  Inadequate interpretation of data and failure to fully explain the observed phenomena in lines 326-329. Since the authors observed significant differences in the kinetics of copper accumulation by U. compressa in different regions, but did not discuss in depth the underlying biological mechanisms and significance, which increases the experimental uncertainty as well as the uncertainty of future applications, the authors should consider this carefully.

5. In lines 397-405, the collection of U. compressa from the Cocholgüe region of Southern Chile is described. However, no information is provided on the specific time of sampling, the specific environmental conditions of the site, or the sampling tools. These details are critical for reproducibility and validation of experimental results. Environmental conditions at sampling sites (e.g., water temperature, salinity, contaminant concentrations, etc.) may have a significant impact on the results of the study and should therefore be documented in detail.

6. In lines 473-475, the use of two-way analysis of variance (ANOVA) and Tukey's multiple comparison test to determine significant differences is only mentioned, but the specific statistical software and version are not described, nor are details on how the data were handled (e.g., normality tests for the data, chi-square tests for variance, etc.). In addition, there is a lack of description of specific significance levels (e.g., p-values specifically set), which are very important parts of statistical analysis.

Author Response

1. There is a problem with the structure of the article, why the materials and methods are placed at the end please explain (usually after the introduction) and a conclusion should be added to summarize your findings.

In IJMS, Mat & Met are placed after Discussion. A sentence of conclusions is highlighted in red in lines 278-281.

  1. The introduction needs to set out more clearly the background and significance of the study, which is currently too long and partly not directly related to the subject of the study.

A paragraph was deleted in introduction to shorten this section.

  1. Some of the results lack detailed data presentation and statistical analysis to adequately support the conclusions drawn, and in lines 285-295, while an increase in copper and sulfides is mentioned, no specific data points or statistical results are provided to demonstrate this change.

A figure (Fig. 7) was added showing the number of electrondense particles in the cytoplasm and the chloroplast in control and treated cells. A sentence was added to the text and highlighted in red.

  1. Inadequate interpretation of data and failure to fully explain the observed phenomena in lines 326-329. Since the authors observed significant differences in the kinetics of copper accumulation by U. compressa in different regions but did not discuss in depth the underlying biological mechanisms and significance, which increases the experimental uncertainty as well as the uncertainty of future applications, the authors should consider this carefully.

A sentence was added to discussion explaining the different amount of copper in juvenile and adult algae (lines 317-320).

  1. In lines 397-405, the collection of U. compressa from the Cocholgüe region of Southern Chile is described. However, no information is provided on the specific time of sampling, the specific environmental conditions of the site, or the sampling tools. These details are critical for reproducibility and validation of experimental results. Environmental conditions at sampling sites (e.g., water temperature, salinity, contaminant concentrations, etc.) may have a significant impact on the results of the study and should therefore be documented in detail.

A sentence was added to materials and methods indicating the conditions of the sampling site and seawater (line 397-398).

  1. In lines 473-475, the use of two-way analysis of variance (ANOVA) and Tukey's multiple comparison test to determine significant differences is only mentioned, but the specific statistical software and version are not described, nor are details on how the data were handled (e.g., normality tests for the data, chi-square tests for variance, etc.). In addition, there is a lack of description of specific significance levels (e.g., p-values specifically set), which are very important parts of statistical analysis.

A sentence mentioning the statistical software and normalization test was added (line 467).

Reviewer 2 Report

Comments and Suggestions for Authors

Reviewers comments on article Copper is accumulated as copper sulfide particles, and not bound to glutathione, phytochelatins or metallothioneins, in 3 the marine alga Ulva compressa (Chlorophyta)

General comments:

Line 39: please mention the main role of GSH and PCs in plants.

Line 115-124: This paragraph suits better R&D section than introduction.

Also please order organisms that you mention in the introduction, so that plants go with plants, yeasts and fungi go apart and so on. Please reorganize intro accordingly.

Line 131: Please state here the novelty of your work clearly.

Line 158 and simmilar: Please avoid this phrase and add the data in appendix.

Author Response

Line 39: please mention the main role of GSH and PCs in plants.

We added a sentence in the introduction that is highlighted in red (lines 58-59).

Line 115-124: This paragraph suits better R&D section than introduction.

The paragraph was transferred to discussion (line 369-373).

Also please order organisms that you mention in the introduction, so that plants go with plants, yeasts and fungi go apart and so on. Please reorganize intro accordingly.

The paragraphs were reorganized.

Line 131: Please state here the novelty of your work clearly.

A sentence was added at the end of introduction regarding the novelty.

Line 158 and similar: Please avoid this phrase and add the data in appendix.

The sentence in line 158 was deleted.
